# Lipid Profile Quantification and Species Discrimination of Pine Seeds through NIR Spectroscopy: A Feasibility Study

**DOI:** 10.3390/foods11233939

**Published:** 2022-12-06

**Authors:** Mariem Khouja, Ricardo N. M. J. Páscoa, Diana Melo, Anabela S. G. Costa, M. Antónia Nunes, Abdelhamid Khaldi, Chokri Messaoud, M. Beatriz P. P. Oliveira, Rita C. Alves

**Affiliations:** 1Laboratory of Nanobiotechnology and Valorization of Medicinal Phytoresources, Department of Biology, National Institute of Applied Science and Technology, University of Carthage, B.P. 676, Tunis Cedex 1080, Tunisia; 2National Research Institute of Rural Engineering, Water and Forests, University of Carthage, B.P. 10, Ariana, Tunis 2080, Tunisia; 3REQUIMTE/LAQV, Department of Chemical Sciences, Faculty of Pharmacy, University of Porto, Rua de Jorge Viterbo Ferreira 228, 4050-313 Porto, Portugal

**Keywords:** pinion, lipids, infrared spectroscopy, chemometrics, PLS, PLS-DA

## Abstract

Pine seeds are known for their richness in lipid compounds and other healthy substances. However, the reference procedures that are commonly applied for their analysis are quite laborious, time-consuming, and expensive. Therefore, it is important to develop rapid, accurate, multi-parametric, cost-effective and, essentially, environmentally friendly analytical techniques that are easily implemented at an industrial scale. The viability of using near-infrared (NIR) spectroscopy to analyse the seed lipid content and profile of three different pine species (*Pinus halepensis*, *Pinus brutia* and *Pinus pinaster*) was investigated. Moreover, species discrimination using NIR was also attempted. Different chemometric models, namely partial least squares (PLS) regression, for lipid analysis, and partial least square—discriminant analysis (PLS-DA), for pine species discrimination, were applied. In relation to the discrimination of pine seed species, a total of 90.5% of correct classification rates were obtained. Regarding the quantification models, most of the compounds assessed yielded determination coefficients (R^2^_P_) higher than 0.80. The best PLS models were obtained for total fat, vitamin E, saturated and monounsaturated fatty acids, C20:2, C20:1n9, C20, C18:2n6c, C18:1n9c, C18 and C16:1. Globally, the obtained results demonstrated that NIR spectroscopy is a suitable analytical technique for lipid analysis and species discrimination of pine seeds.

## 1. Introduction

The *Pinus* genus comprises about 250 species, mainly found in the Northern hemisphere. Aleppo pine (*Pinus halepensis* Mill.) and Maritime pine (*Pinus pinaster* Ait.) are among the most important forest species in the Mediterranean Basin [1]. Pine nuts are considered a delicacy and, for that reason, are commonly added to foods such as salads, breads, vegetables, and meat [2] For example, in Tunisia, Aleppo pine seeds are used as an ingredient in ice-creams, candies and also to prepare a traditionally sweet pudding [3]. Pine seeds are also known to possess an important fat content [4,5] and due to their antioxidants content [6] and their richness in polyunsaturated, monounsaturated fatty acids (especially linoleic acid and oleic acid) and vitamin E [4,5,7], can play an important role in human nutrition and health. Moreover, the neutral lipids, glycolipids, and phospholipids of pine seeds were also related to antiangiogenic activity [8]. In this sense, it is important to have an analytical tool capable of assessing their lipid content and if possible, to discriminate pine seeds species based on their different compositions [9]. This analytical tool should be rapid, accurate, multi-parametric, cost-effective, and environmentally friendly.

On the contrary, the traditional chemical methods usually applied for assessing the lipid content are small-scale, time-consuming, labor-intensive, costly, and laboratory dependent. Indeed, the determination of fatty acids content is mainly performed by gas chromatography, H NMR and C NMR [10,11]. Thus, bearing in mind all the drawbacks abovementioned related with these techniques, the development of other analytical methods capable of providing reliable results without these disadvantages are necessary.

Vibrational spectroscopic techniques possess all the desired characteristics and near-infrared (NIR) spectroscopy constitutes an interesting alternative. NIR spectroscopy results from complex overtones and combination bands of fundamental vibrations from C-H, N-H, O-H and S-H bonds within the spectral range of 14,000 to 4000 cm^−1^ [12]. However, as this information arises into a complex spectrum, the use of chemometrics is essential to extract useful information.

In fact, the use of vibrational spectroscopic techniques regarding authentication purposes has already been explored [13,14,15]. More specifically, considering only NIR spectroscopy with a similar purpose than the one proposed hereby, fatty acid composition [16,17] and species discrimination [15] have already been presented with satisfactory results. Therefore, the application of NIR spectroscopy for lipids quantification and species discrimination in pine seeds seems to be an interesting approach. As far as we know, few works have been carried out to study the application of this technique in pine. Most of the existing works have been applied to pine wood. In relation to the application of NIR spectroscopy to pine seeds, only the following works have been published [18,19,20,21].

In the first published work, NIR spectroscopy was used to predict the moisture content in seeds. Good predictive models using partial least squares (PLS) regression were obtained with correlation coefficients and prediction errors for the test sets of 0.99 and lower than 2%, respectively [20]. Two years later, the same authors applied NIR spectroscopy for moisture determination and the developed model yielded prediction errors around 1.1% [21]. A few years later, NIR spectroscopy was used for discriminating viable, dead-filled and empty pine seeds of three different pine species. The obtained results through orthogonal projection to a latent structure-discriminant analysis (OPLS-DA) model yielded good results, with a sensitivity and specificity of 100% and 99%, respectively [19]. In a similar work to the one herein described, NIR spectroscopy was applied for the determination of total lipid content in pine seeds and also their origin, using only one pine specie (*Pinus koariensis*). Both the determination of total lipid content and the origin of the pine seeds yielded good results [18].

As far as we know, there have been no previous studies on the discrimination of Tunisian pine seeds through NIR spectroscopy. Accordingly, the aim of this work was to determine the lipid profile and discriminate the seeds of three different pine species (*Pinus halepensis*, *Pinus brutia* and *Pinus pinaster*) using NIR spectroscopy.

## 2. Materials and Methods

### 2.1. Preparation of Samples

Twenty samples of pine seeds were composed of sixteen populations of Aleppo pine (*P. halepensis*), two populations of Maritime pine (*P. pinaster*) and two of Brutia pine (*P. brutia*) belonging to different geographic and bio-climatic zones in Tunisia (Table 1) collected in 2018. Before analysis, pine seeds were ground in a mill (GM Grindomix 200, Retsh, Haan, Germany). Different parameters were determined to characterize pine seeds.

### 2.2. Lipid Content and Lipid Profile Analysis

Total fat content was determined by Soxhlet [22]. The vitamin E content was analysed by high performance liquid chromatography (HPLC) system after extraction of lipid fraction according to [23]. For the analysis of fatty acids, the preparation of fatty acids methyl esters (FAMEs) was made according to [24] with minor modifications in order to identify the profile using gas chromatography-flame ionization detection (GC-FID).

#### 2.2.1. Total Lipids Quantification

Total lipid content was determined by extracting one gram of crushed seeds with 100 mL petroleum ether in a Soxhlet apparatus for 8 h. The remaining solvent was recovered and the residue was dried at 103 ± 2 °C till constant weight [22].

#### 2.2.2. Lipid Profile Analysis

The lipid fraction was obtained by a cold extraction according to Alves et al. [23]. An amount of sample (70 mg), 75 μL of BHT 0.1%, 50 μL of tocol (used as internal standard at 0.1 mg/mL) and 1 mL of absolute ethanol were mixed with a mechanical homogenization during 30 min in an orbital vortex mixer (VV3, VWR Int., Porto, Portugal). A second homogenization for 30 min was made after addition of 2 mL of hexane and the mixture was left overnight at 4 °C. Then, 1 mL of NaCl 1% (*m*/*v*) was added. After centrifugation (5000× *g* rpm/5 min), the organic layer was collected, and a second extraction was made with the addition of 2 mL of hexane. The organic phases were combined, and an amount of anhydrous sodium sulfate was mixed with the extract to eliminate the traces of water. After a new centrifugation (13,000× *g* rpm, 10 min), the supernatant was collected and concentrated under a nitrogen steam until 1 mL.

For Vitamin E profile, an aliquot of the extract prepared in the Section 2.2.2. was transferred to an amber vial and injected into a normal phase HPLC system according to Alves et al. [23]. The analysis was performed in triplicate. The identification of compounds was made by their UV spectra and their retention times compared to standards, and total vitamin E amount were obtained from the sum of the individual vitamers.

For fatty acids, another aliquot of the extract obtained in Section 2.2.2. was used to prepare fatty acids methyl esters (FAMEs) according to Costa et al. [24] with minor modifications. After the evaporation of n-hexane, 1 mL of dichloromethane and 2 mL KOH (0.5 M in methanol) were added to the residue (~15–20 mg of lipids). The mixture was vortexed and heated at 90 °C for 10 min in a thermo block (SBH130D/3, Stuart, Stafford, UK) and 2 mL of boron trifluoride solution (14% in methanol) were added, after cooling with ice for 5 min. The mixture was heated at 90 °C for 30 min and placed to cooling again. A volume of 2 mL of deionized water and n-hexane were added. After homogenization and centrifugation at 3000× *g* rpm for 5 min., the organic layer was transferred to another vial. Anhydrous sodium sulfate was used to eliminate any remaining water and 1 mL of the supernatant was transferred into a 2 mL vial after centrifugation to be analyzed by GC-FID according to Costa et al. (2018) [24]. FAMEs were identified by comparison of the retention times with those of the standard mixture Supelco 37 Component FAME Mix (Bellefonte, PA, USA). Results were expressed as relative percentages of each fatty acid.

### 2.3. Acquisition of Near-Infrared Spectra

The ground pine seed samples (around 1.5–2 g) were transferred to borosilicate flasks before spectral acquisition. The acquisition of NIR spectra was made on reflectance mode using a Fourier transform near infrared spectrometer (FTLA 2000, ABB, Quebec, Canada) equipped with a detector of indium-gallium-arsenide (InGaAs). The equipment was controlled by Bomen-Grams software (version 7, ABB, Canada). The spectra were acquired within 10,000 to 4000 cm^−1^, with a resolution of 8 cm^−1^ and an average of 64 scans. Each sample was analysed in triplicate. A Teflon reference material was used for the background before the spectral acquisition of pine seed samples (just one background was needed as the spectral acquisition lasted around 1 h without signal deterioration).

### 2.4. Data Analysis

Three different chemometric tools, namely principal component analysis (PCA) [25], partial least squares (PLS) [26] regression and partial least squares-discriminant analysis (PLS-DA) [27], were used. PCA was used to assist the detection of outliers through squared residuals statistics and Hotelling′s T^2^, and to find common tendencies. PLS was selected for the development of quantification models while PLS-DA was chosen for pine seeds species discrimination using NIR spectra and lipid profile chemical values. Before application of PCA, PLS and PLS-DA, the NIR data were mean centered. All the calculations were performed through Matlab R2014a version 8.3 (MathWorks, Natick, MA, USA) using PLS Toolbox version 8.2.1 (Eigenvector Research Inc., Wenatchee, WA, USA).

The spectra were divided into two different groups for PLS and PLS-DA. The first group encompassing 70% of the samples was used for calibration and the respective optimization. The second group encompassing 30% of the samples was used for validation purposes. The division was performed randomly but ensuring that the values of the validation set were within the values of the calibration set for PLS and the same proportion of samples in all groups to prevent unbalanced groups for PLS-DA. Therefore, a total of 33 spectra of Aleppo pine, four spectra of Maritime pine and four spectra of Brutia pine were used in the calibration and the rest for validation (15 spectra of Aleppo pine, two spectra of Maritime pine and two spectra of Brutia pine). The optimization of the PLS and PLS-DA models involved studying different spectral regions, pre-processing techniques and estimating the best number of latent variables (LV) using only the calibration set. The assessment of the best number of latent variables was performed through the leave-one-sample-out cross validation method to prevent model over-fitting. The NIR spectra were divided into five different spectral regions (spectral region R1 from 4961 to 4016 cm^−1^, spectral region R2 from 5423 to 4964 cm^−1^, spectral region R3 from 6079 to 5427 cm^−1^, spectral region R4 from 7776 to 6083 cm^−1^ and spectral region R5 from 9975 to 7780 cm^−1^) and all these spectral regions were tested individually or in all possible combinations. The pre-processing techniques tested were standard normal variate (SNV) and Savitzky–Golay filter (using the first and second derivative orders, different filter widths and second polynomial order), individually and in all possible combinations. The best PLS calibration models were selected based on the lowest root mean square error of calibration (RMSEC) and root mean square error of cross-validation (RMSECV) while the PLS-DA calibration models were selected based on highest percentage of correct predictions. After the optimization of the PLS and PLS-DA calibration models, the validation set was projected to obtain the correct predictions percentage through PLS-DA and to obtain the quantification results through PLS. The obtained results through PLS-DA were compiled in a confusion matrix format where the sum of the diagonal elements provides the total percentage of correct predictions. The obtained results through PLS were evaluated using different parameters as, root mean square error of prediction (RMSEP), coefficient of determination of prediction (R^2^_P_) and range error ratio (RER). The RMSEC, RMSECV and prediction RMSEP were calculated according to the following equation:(1)$$RMSE=\sqrt{\frac{\sum_{i=1}^{n}{\left(c_{i}-{\hat{c}}_{i}\right)}^{2}}{n}}$$
where, n is the number of samples, $c_{i}$ is the experimental measurement for sample i and ${\hat{c}}_{i}$ is the corresponding value obtained for calibration (RMSEC), cross-validation (RMSECV) and prediction set (RMSEP).

The RER parameter was calculated using the following equation:(2)$$RER=\frac{\left(Y_{max}-Y_{min}\right)}{RMSEP}$$

For the PLS-DA applied to the lipid profile chemical values, the same strategy mentioned before for data division was followed and these data were firstly autoscaled and then mean centered. No other pre-processing techniques were tested. Note that each sample was analysed in triplicate through the reference procedures and all the values were used instead of considering the average. Again, the obtained results were compiled in a confusion matrix format where the sum of the diagonal element provides the total percentage of correct predictions.

## 3. Results and Discussion

### 3.1. Total Lipid Content and Lipid Profile of the Seeds

All the paremeters analysed as well as their maximum and minimum values are given in Table 2.

Total lipid content varied between 13.6 (*P. pinaster*) to 33.7% (*P. halepensis*). The differences were explained by the difference of species; the degree of seeds maturation, exposure to different climatic conditions differences, and by phenotypic and genotypic diversities among populations and species [2,26,27,28].

For fatty acids, the seeds lipid fractions were characterized by their richness in polyunsaturated (61.0 to 66.2%) and monounsaturated (22.8 to 27.9%) fatty acids where the major compounds are linoleic and oleic acids (60.6–64.8 and 21.7–27.1%, respectively). Our results are in agreement with those reported in the literature, where unsaturated fatty acids had the major percentage of the composition of seed oils (more than 90%), with oleic and linoleic acids present as the main compounds [2,3,4,7].

The seeds of pine species were also characterized by a higher quantity of vitamin E, ranging from 125 (*Pinus pinaster*) to 260 mg/Kg of seeds, where those of *P. halepensis* contained the uppermost content. The vitamin E contents determined in this study are higher than those determined by Cheikh-Rouhou et al. [5] and Dhibi et al. [4] for Tunisian *P. halepensis* seeds (157 and 178.9 mg/kg of oil, respectively). Matthaus et al. [2] studied the difference between 22 species of pine and the value of vitamin E content varied between 107 and 667 mg/Kg of oil.

### 3.2. NIR Spectra

The raw NIR spectra of pine seeds are depicted in Figure 1. It can be observed that the most prominent bands excluding the bands in water regions (spectral regions R2 and R4) are found around 8300, 5840, 5700, 4860, 4640, 4340 and 4255 cm^−1^. The spectral region around 8300 and 5840 and 5700 cm^−1^ are related with the C-H bond of the second overtone and C-H bond of the first overtone, respectively. The bands around 4860 and 4640 cm^−1^ may be related to proteins (N-H) and oils (C-H stretch), respectively [28]. Moreover, pine seeds were characterized by their richness in protein with a percentage reaching 22.7 to 33.6 g/100 g [3] and from 14% to 27% [1] depending on the species for total protein 13.0 to 41.9 g/100 g for total amino acids [2]. The bands around 4340 and 4255 cm^−1^ with oils (C-H bend of second overtone) and cellulose (C-H bend of second overtone), respectively [28].

### 3.3. PCA Analysis

All the obtained spectra were used because no outliers were detected in PCA through the analysis of squared residuals statistics and Hotelling′s T^2^. PCA was also used to verify if pine seeds clustered according to its species. Figure 2 shows the scores obtained from PC1 (90.1% of variance) versus PC3 (1.43% of variance) with the NIR spectra of pine seeds mean centered and considering all spectra. A slightly clustering tendency is visible in relation to the species analyzed. In fact, Maritime pine samples tend to have negative scores in PC1 and positive scores in PC3, Brutia pine samples tend to have negative scores in both PC1 and PC3 and Aleppo pine samples shows a tendency for having positive values in PC1. These results suggest the application of a supervised method, as PLS-DA, for the discrimination of pine seeds species because PCA does not force the model to find any relations. PC3 was shown instead of PC2 (7.42% of variance) as PC1 versus PC3 showed a better cluster tendency in terms of pine seeds species than PC2.

### 3.4. Discrimination Analysis

#### 3.4.1. Using NIR Spectra

As abovementioned, several PLS-DA calibration models were developed to find the best spectral region and pre-processing technique for the discrimination of pine seeds species. The highest percentage of correct predictions was found when using the combination of spectral regions R1 and R3 and with the NIR spectra mean centered. With these conditions a total of 94.7% (18/19) of correct predictions were obtained using 2 LV. Note that for assessment of the correct predictions, only the validation set was used. The respective confusion matrix is shown in Table 3.

Besides providing the total amount of correct predictions, the confusion matrices also revealed the species that were the best and worst predicted. The Maritime and Brutia pine species were the best-predicted species with a total of 100% of correct predictions. The worst results were obtained with the prediction of Aleppo pine species where one validation sample was confused with Brutia pine species. Nonetheless, around 93% of correct predictions were obtained for Aleppo pine specie. These findings agree with the obtained results with PCA (Figure 2) where it can be observed that the Brutia pine species clustered closely to Aleppo pine species. The results obtained with PCA and PLS-DA seem to indicate that these species are very similar in their NIR spectra. Nonetheless, the obtained results (around 95% of total correct predictions) attest the suitability for discriminating pine seeds species with NIR spectroscopy. With the objective of understanding the more important spectral regions for the developed PLS-DA, the respective regression coefficient vectors were depicted in Figure 3.

The wavenumbers that showed the highest contribution to the best PLS-DA model were located around 6070, 4340 and 4260 cm^−1^. The wavenumbers of 4340 and 4260 cm^−1^ can be ascribed with the oil (C-H bend second overtone) and cellulose (CH_2_ bend second overtone) content, respectively [28]. The wavenumbers around 6070 cm^−1^ may be related to compounds that contain C-H bonds. In our opinion, these findings could make sense considering that it is probable that the amount of oil and cellulose could differ within pine seeds species. In fact, Ovcharova and co-authors [29] have already mentioned that the composition of grape seeds can be affected to some degree by the grape variety. However, additional studies are needed to confirm that the discrimination found due to the C-H bonds are related to the different pine seed species or the climate conditions or geographical area.

#### 3.4.2. Using Lipid Profile Chemical Values

As the obtained results for the discrimination of pine seeds species through PLS-DA applied to NIR spectra suggested that the oil content may be responsible for this discrimination, the chemical values obtained for the lipid profile were tested for the same purpose. In this case, as mentioned in the Section 2.4, the chemical values were only autoscaled and then mean centered. The highest number of correct predictions using only the calibration set were obtained with 2 LVs. Then, the validation set was then projected and a total of 94.7% (18/19) of correct predictions were obtained using 2 LV. Again, note that for assessment of the correct predictions, only the validation set was used. The respective confusion matrix is given in Table 4.

Again, the best predictions were obtained for Maritime and Brutia pine species and the worst prediction involved Aleppo pine species which was slightly misclassified as Brutia pine (6.7%) species, suggesting that the lipid profile of both these species are very similar. Nonetheless, around 93% of correct predictions were obtained for Aleppo pine species. Once more, these results are in agreement with the obtained results with PCA (Figure 2) and PLS-DA when using NIR spectra.

The value for the total percentage of correct predictions through the lipid profile chemical values is the same as the one obtained considering NIR spectra and reveals that pine seeds can be discriminated regarding its lipid content using their lipid profile chemical values. This was expected, as the NIR spectra, in theory, gathers more information regarding the chemical composition of pine seeds than just their lipid content.

The respective coefficient vectors for this PLS-DA were analysed (data not shown) and the most important lipid parameter for this discrimination was C14:0.

### 3.5. Quantification Analysis

After the discrimination analysis regarding pine seeds species which revealed interesting results, another objective of this work was the application of NIR spectroscopy for the lipid profile quantification of pine seeds samples. The parameters evaluated as well as their maximum and minimum values are shown in Table 2. Several PLS models using only the calibration set were developed aiming to find the best spectral regions and pre-processing techniques as well as the optimal number of latent variables. After finding the best PLS calibration models the independent test samples were projected to test the accuracy of the developed models. The obtained results for the best calibration models regarding each assessed parameter are shown in Table 5. The best spectral regions for most of the parameters assessed were spectral regions R1 and R3. This was expected as these regions are the most informative ones (Figure 1) and can be related with oils content [28]. In relation to the best pre-processing technique, both the application of Savitzky-Golay filter (using the first or second order derivative) followed by SNV or just SNV alone yielded the best results. The application of SNV removes the scatter effects that are common using diffuse reflectance spectroscopy while the first and second derivatives help to remove the additive and linear baseline, respectively [30].

The obtained R^2^_P_ for most of the compounds assessed were higher than 0.75 which indicates that is possible to quantify the amount of lipids through the NIR spectra of pine seeds. The best PLS models were obtained for total fat, vitamin E, saturated and monounsaturated fatty acids, C20:2, C20:1n9, C20, C18:2n6c, C18:1n9c, C18 and C16:1. These results are in agreement with the findings of Galtier et al. (2007) [31] who also established accurate PLS models for the prediction of the major FAs with excellent prediction abilities for C16:1 (R^2^ = 0.85), C18:1 (R^2^ = 0.97), C18:2 (R^2^ = 0.98), C20:1 (R^2^ = 0.88). For saturated and monounsaturated fatty acids, our results are well in line with the findings of Mossoba et al. (2013) [32] who reported PLS-R models with a values of R^2^ equal to 0.9957 and 0.9993, respectively.

It is true that the developed PLS models are based in a low number of samples but the obtained results clearly indicate that it is possible to obtain a lipid profile of pine seeds. More samples are needed to attest the reliability of the developed models, nonetheless the obtained results were very positive as a proof of concept.

The projection of the validation set into the PLS calibration models for the parameters that yielded good results are shown in Figure 4 and Figure 5. It can be seen that some of these PLS models presented a low range of values, which together with a low number of samples, has a negative impact on the obtained results. Therefore, further studies (including a higher number of samples in a wider range of values) are needed to attest the accuracy and robustness of the models.

The regression coefficient vectors for the PLS models shown in Figure 4 and Figure 5, were depicted in Figure 6 and Figure 7 and analysed to find the most important wavenumbers to each respective PLS model.

For total fats, the most important wavenumbers were located within 5970 to 5400 cm^−1^, and within 4400 to 4200 cm^−1^. The former interval can be related to the content of fatty acids [33] and cellulose [28], while the later interval can be connected with the amount of oil, protein and cellulose [28]. The wavenumbers that were more relevant for vitamin E were observed within 5450 to 5400 cm^−1^ and around 5520 cm^−1^, which can all be associated with fatty acids [33,34]. There is no clear evidence of a relation between fatty acids and vitamin E content but Goffman and Böhme [35] reported a moderate positive correlation (r = 0.41) between vitamin E equivalents and polyunsaturated fatty acids in maize hybrids. For C18:1n9c PLS models, the wavenumbers with the highest contribution were located around 6000 to 5500 cm^−1^ which can be correlated with fatty acids [33]. For the PLS model of saturated fatty acids the wavenumbers around 4420 and 4050 cm^−1^ and within 5950 to 5700 cm^−1^ were the most significant ones. The former wavenumbers can be related with the cellulose and protein [28] while the later region can be connected with the amount of fatty acids [33]. Regarding monounsaturated fatty acids, the more important wavenumbers were located around 7200 cm^−1^ and within 5950 to 5750 cm^−1^. The former wavenumber can be associated with C-H bonds [28,36] while the latter wavenumber region can correlated with fatty acids [35]. The C-H bonds can be found in many different compounds, namely in oils, but none of the existing literature refer them for the determination of oils [28,33,34,36]. Further studies are needed to elucidate this. In relation to C20:2, the most relevant wavenumbers were observed around 5950, 5800, 5570, 5350 and 5250 cm^−1^. The first three wavenumbers can be related with fatty acids [33] while the last two can be connected with C=O bonds [28] which can be found in several compounds including oils. Again, further studies are needed to clarify this. For C20:1*n*9, the most important wavenumbers were found around 6000 cm^−1^ and within 4600 to 4300 cm^−1^ and 4150 to 400 cm^−1^. The wavenumbers around 6000 cm^−1^ may be associated with fatty acids [33] while the former wavenumber interval can be correlated with fatty acids, proteins, cellulose and oil [28,34]. The latter wavenumber interval can be connected with proteins and cellulose [28]. Regarding C20:0, the most significant wavenumbers were located around 9950, 8250, 8100 and 7800 cm^−1^. These wavenumbers can be related with C-H, N-H, C=O and O-H bonds which can associated with different compounds, namely oils. Once again, further studies are needed as none of the existing literature [28,33,34,36] connects them with oils determination. In relation to C18:3*n3*, the wavenumbers within 5900 and 5750 cm^−1^ that can be correlated with fatty acids content [33] were the most relevant ones. Regarding C18:0, the most important wavenumbers were located within 5250 to 5200, 4600 to 4380 and 4100 to 4000 cm^−1^. These wavenumbers can be related with the amount of proteins and cellulose [28]. Finally, for C22:0 the wavenumbers around 5975, 5810, 5400 and 4980 cm^−1^ were the ones with the highest contribution to the PLS model. The fatty acids content can be associated with the first two wavenumber regions [33], while the wavenumbers around 5400 and 4980 cm^−1^ may be connected with cellulose [28] and amine [34], respectively.

Globally, most of the more important wavenumbers identified in the regression coefficient vectors for all the PLs models were related with the amount of fatty acids which reinforces that it is possible to quantify the lipid content in pine seeds even with a low number of samples. Some regression coefficient vectors presented a noisy feature that can be associated with the use of spectral region R5. Although the optimization of the PLS models indicated this spectral region was important, it is the less informative and noisier spectral region.

It is true that more samples are needed to attest the robustness of the developed PLS models. However, as a proof of concept, the obtained results by PLS (R^2^_P_, RMSEP) and PLS-DA (total percentage of correct predictions) as well as the respective regression coefficient vectors, clearly indicate that NIR spectroscopy can be applied for the determination of lipids in pine seeds and the discrimination of species.

## 4. Conclusions

The determination of lipids in pine seeds is very important due to their rich composition in mono- and polyunsaturated, namely fat content, that are related with several benefits for human health.

This manuscript described an environmentally friendly, rapid, cost-effective, multi-parametric and accurate analytical technique based on NIR spectroscopy for the determination of lipids in pine seeds as well as for the discrimination of the different species included in this study.

The obtained results through PLS-DA (when using NIR spectra or lipid profile chemical values) and PLS were very satisfactory despite the low number of samples. In fact, around 95% (18 samples from the 19 samples of the validation set) of correct classification rates were obtained for the discrimination of these three pine seeds species considering the NIR spectra and lipid profile chemical values. The regression coefficient vectors obtained when using NIR spectra indicated that the discrimination obtained can be connected with the different amount of oils and cellulose. Regarding lipid profile chemical values, the regression coefficient vector suggested that the most important chemical parameter was C14:0.

For PLS, the obtained R^2^_P_ for most of the compounds assessed were higher than 0.75 which indicates the suitability of NIR spectroscopy. The best PLS models were obtained for total fat, vitamin E, saturated and monounsaturated fatty acids, C20:2, C20:1n9, C20, C18:3n3, C18:1n9c, C18 and C22:0. The analysis of the regression coefficient vectors of the best PLS models revealed that the most important wavenumbers identified could be associated with the amount of fatty acids in pine seeds. These findings reinforce the results of developed PLS models and also the suitability of NIR spectroscopy for the quantification of lipids in pine seeds.

Further studies, including a higher number of samples for each species over a wider range of the studied parameters, are needed to confirm the robustness of this methodology. In more detail, this should include samples: from the same species from the same region as well as from different regions; from different countries; under the same and different agricultural practices; at different maturation stages; and, if possible, over different years. The precautions that should be considered when using this method are related to the moisture level and granulometry of the samples, as these factors affect the NIR spectra. Generally, the primary results obtained using NIR spectroscopy coupled with PLS and PLS-DA suggest that this technique could be applied for the quantification of lipids and the discrimination of pine seeds species.

## Figures and Tables

**Figure 1 foods-11-03939-f001:**
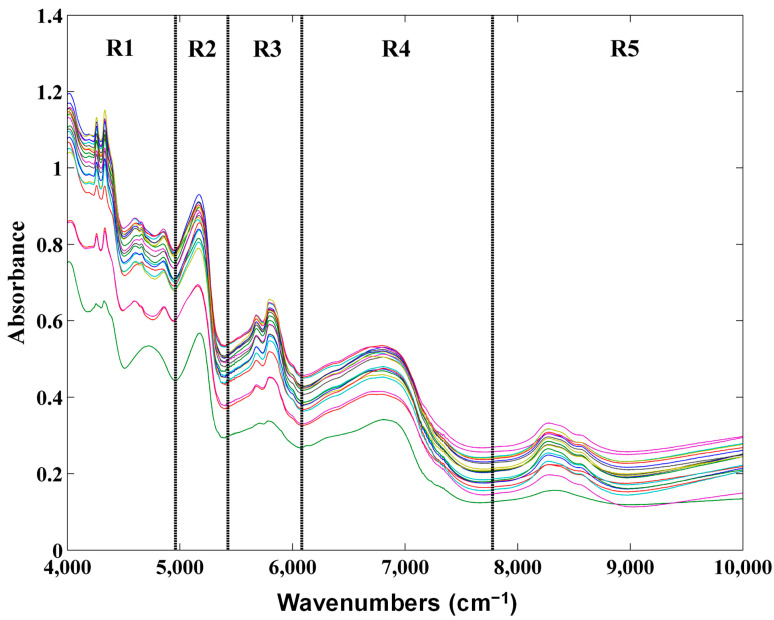
Raw NIR spectra of pine seeds.

**Figure 2 foods-11-03939-f002:**
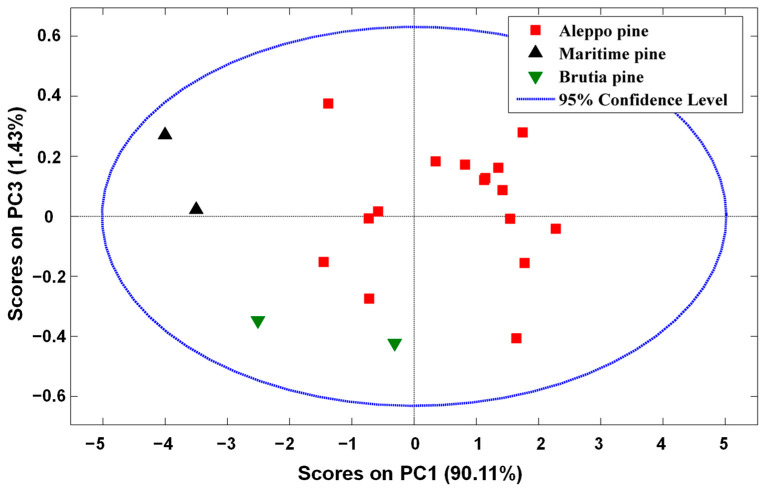
Scores obtained of PC1 versus scores obtained of PC3 with NIR spectra mean centered and considering all spectra.

**Figure 3 foods-11-03939-f003:**
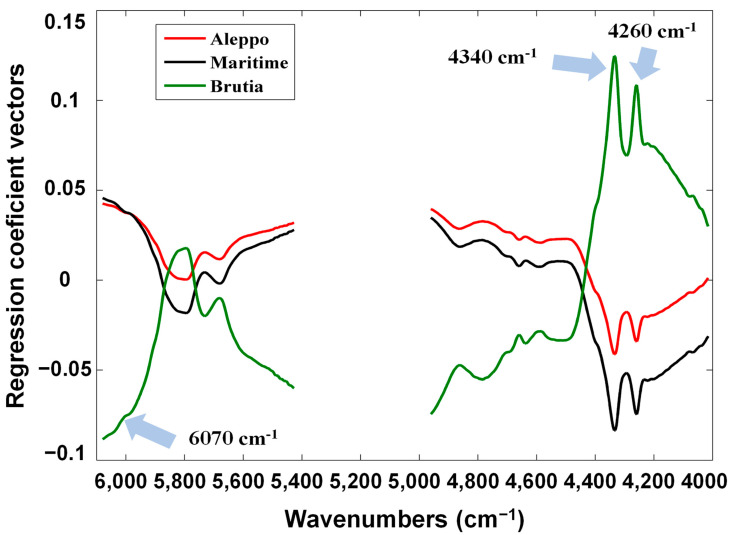
Regression coefficient vectors for the best PLS-DA model regarding pine seeds species discrimination using the spectral regions R1 and R3 and 2 LV for the validation set.

**Figure 4 foods-11-03939-f004:**
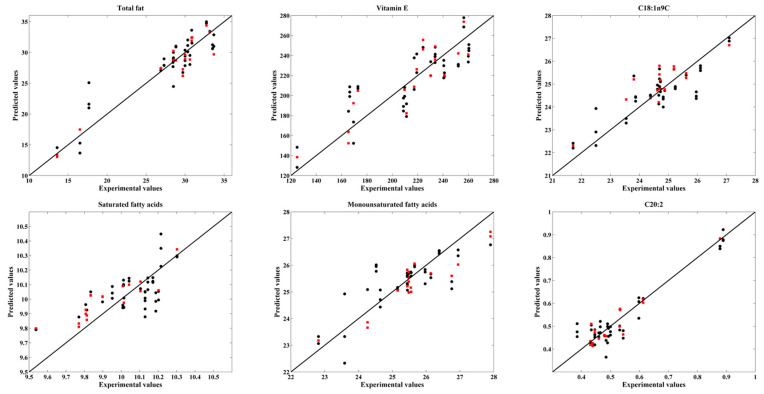
Experimental values versus the cross-validation (●) and validation (

) values obtained for total fat, vitamin E, C18:1n9c, saturated fatty acids, monounsaturated fatty acids and C20:2.

**Figure 5 foods-11-03939-f005:**
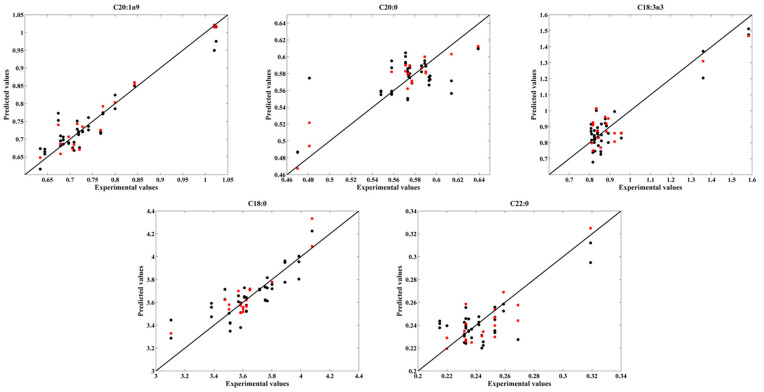
Experimental values versus the cross-validation (●) and validation (

) values obtained for fatty acids C20:1n9, C20:0, C18:3n, C18:0 and C22.

**Figure 6 foods-11-03939-f006:**
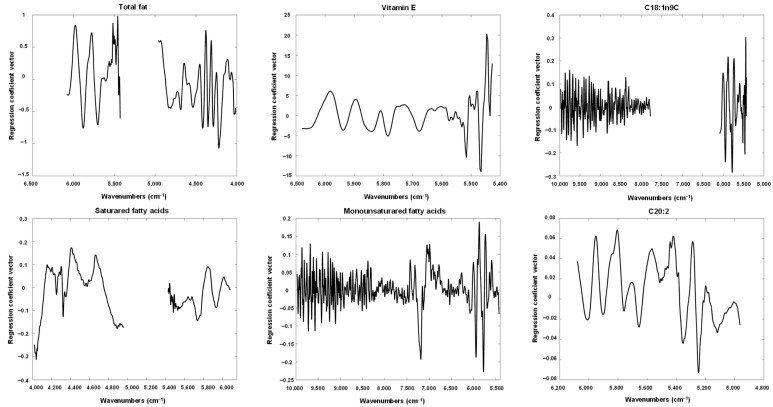
Regression coefficients vectors of the best PLS models built for total fat, vitamin E, C18:1n9c, saturated fatty acids, monounsaturated fatty acids and C20:2.

**Figure 7 foods-11-03939-f007:**
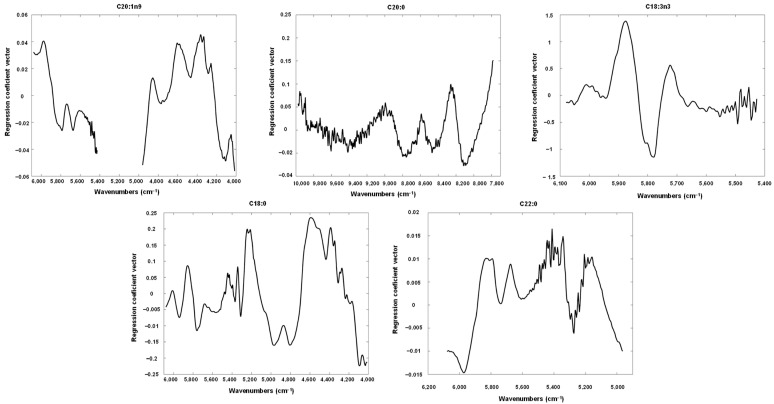
Regression coefficients vectors of the best PLS models built for fatty acids C20:1n9, C20:0, C18:3n, C18:0 and C22.

**Table 1 foods-11-03939-t001:** Geographical coordinates and ecological characteristics (latitude, longitude, altitude, bioclimate, Pluviometry, temperature) of the plant material harvesting sites.

Geographical and Ecological Characteristics	Harvest Site							
	Dar Chichou	Kasserine	Beja	Sousse	Amdoun	Korbous	Henchir Naam	Mjez El Beb
Latitude (N)	36°96′	35°15′	36°42′	35°49′	36°82′	36°50′	36°13′	36°38′
Longitude (E)	10°98′	8°45′	9°10′	10°38′	9°05′	10°35′	9°10′	9°36′
Altitude (m)	39	707	248	25	448	180	450	51
Pluviometry (mm)	504	216	508	354	650	474	509	508
Temperature (°C)	18.5	17.5	19.5	19.4	18	18.6	19	19.6

**Table 2 foods-11-03939-t002:** Maximum and minimum values obtained with the analysed parameters.

Parameter	Minimum Value	Maximum Value	Parameter	Minimum Value	Maximum Value
**Total fat**	13.6	33.7	**C20:0**	0.470	0.639
**C14:0**	0.0406	0.0796	**C20:1*n*9**	0.634	1.02
**C16:0**	5.20	5.64	**C20:2**	0.385	0.889
**C16:1**	0.0552	0.0895	**C22:0**	0.215	0.319
**C17:0**	0.0607	0.0770	**C24:0**	0.480	0.0848
**C18:0**	3.10	4.08	**SFA**	9.54	10.3
**C18 1n9c**	21.7	27.1	**MUFA**	22.8	27.9
**C18:2*n*6c**	60.6	64.8	**PUFA**	61.0	66.2
**C18:3*n*3**	0.81	1.58	**Vitamin E**	125	260

SFA, saturated fatty acids; MUFA, monounsaturated fatty acids; PUFA, polyunsaturated fatty acids. Total fat expressed in g/100 g; Fatty acids expressed in relative %; Vitamin E expressed in mg/Kg of seeds.

**Table 3 foods-11-03939-t003:** Confusion matrix for the best PLS-DA pine seeds species discrimination model based on the spectral regions R1 and R3 of NIR spectra mean centered and using 2 LV considering only the validation data.

Real Pine Seeds Species	Predicted Pine Seeds Species		
	Aleppo	Maritime	Brutia
Aleppo	93.3% (14/15)	0% (0/15)	6.7% (1/15)
Maritime	0% (0/2)	100% (2/2)	0% (0/2)
Brutia	0% (0/2)	0% (0/2)	100% (2/2)

**Table 4 foods-11-03939-t004:** Confusion matrix for the best PLS-DA pine seeds species discrimination model based on lipid profile chemical values autoscaled and then mean centered using 2 LV considering only the validation data.

Real Pine Seeds Species	Predicted Pine Seeds Species		
	Aleppo	Maritime	Brutia
Aleppo	93.3% (14/15)	0% (0/15)	6.7% (1/15)
Maritime	0% (0/2)	10.5% (2/2)	0% (0/2)
Brutia	0% (0/2)	0% (0/2)	10.5% (2/2)

**Table 5 foods-11-03939-t005:** Summary of the best PLS models developed.

Parameter	LV	Best Spectral Region	Best Pre-Processing Technique(s)	RMSEC	RMSECV	RMSEP	R^2^_C_	R^2^_P_	RER
Total fat	5	R1 + R3	SG(15,2,1) + SNV	1.54	2.19	1.70	0.92	0.91	11.8
C14:0	1	R1 + R3	SNV	0.013	0.014	0.011	0.08	0.12	3.7
C16:0	2	R1 + R3	none	0.14	0.15	0.11	0.10	0.03	3.9
C16:1	2	R3 + R5	SG(15,2,2) + SNV	0.0054	0.0064	0.0049	0.57	0.63	6.3
C17:0	2	R3	SG(15,2,1) + SNV	0.0034	0.0038	0.0031	0.49	0.35	5.2
C18:0	6	R1 + R2 + R3	SG(15,2,0) + SNV	0.093	0.12	0.10	0.81	0.80	9.4
C18:1*n*9c	2	R3 + R5	SG(15,2,2) + SNV	0.58	0.68	0.59	0.77	0.75	9.2
C18:2*n*6c	6	R3	SG(15,2,1) + SNV	0.57	0.75	0.66	0.70	0.59	6.5
C18:3*n*3	5	R3	SNV	0.059	0.077	0.085	0.90	0.82	9.1
C20:0	4	R5	SNV	0.014	0.025	0.016	0.88	0.84	10.7
C20:1*n*9	5	R1 + R3	SNV	0.026	0.032	0.027	0.95	0.92	14.5
C20:2	6	R2 + R3	SG(15,2,1) + SNV	0.034	0.048	0.035	0.94	0.90	12.9
C22:0	4	R2 + R3	SNV	0.011	0.015	0.013	0.74	0.73	7.7
C24:0	4	R1	SNV	0.0052	0.0063	0.0061	0.43	0.37	6.0
Saturated fatty acids	4	R1 + R3	SNV	0.10	0.13	0.12	0.76	0.73	6.6
Monounsaturated fatty acids	3	R3 + R4 + R5	SG(15,2,2) + SNV	0.54	0.67	0.55	0.75	0.86	9.4
Polyunsaturated fatty acids	5	R5	SG(15,2,1)	0.43	0.76	0.68	0.89	0.66	6.9
Vitamin E	2	R3	SG(15,2,2) + SNV	17.7	19.7	19.5	0.78	0.76	6.9

Legend: SG—Savitzky-Golay filter (X [filter width], Y [polynomial order], Z [derivative order]); SNV—standard normal variate; the units of RMSEC, RMSECV and RMSEP are the same as those of the reference procedures—please see Table 2.

## Data Availability

The data presented in this study are available on request from the corresponding author. The data are not publicly available due to being used in other future works.

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
