# Peer review of "Lipid Profile Quantification and Species Discrimination of Pine Seeds through NIR Spectroscopy: A Feasibility Study"

_foods, 2022, doi:10.3390/foods11233939_

Round 1

Reviewer 1 Report

GENERAL STATEMENT: The manuscript focuses on the application of NIR spectroscopy and chemical analysis of lipids combined with chemometrics for the identification of the species of three pine species. NIR spectra were also calibrated against the chemical lipid profile to develop quantitative methods able to rapidly quantify these compounds into the pine seeds. The main advantages of proposed methodology over the classical “wet-chemistry”-based ones rely on the possibility of saving time and costs associated with analysis and eliminating the consumption of samples and solvents. Nevertheless, the small sample size is an important flaw which prevent the proper evaluation of some of the statistical tool applied. Some important revisions need to be addressed before the manuscript is considered for publication.

Since the manuscript has been provided without continuous line numbers and pages, revisions have been provided directly within the submitted PDF file as comments to the text. Please refer to the enclosed file.

Author Response

Introduction section

Please check grammar and spelling

Reply: The authors acknowledge the reviewer comment. The sentence was rephrased as suggested.

The novelty of the work and the advances compared to previously published data must be better highlighted in this section.

Reply: The authors acknowledge the reviewer comment. As far as we know, there are no previous studies on the discrimination of Tunisian pine seeds through NIR spectroscopy, so this topic is completely new. Accordingly, that information was added to the manuscript.

Line 93: “As far as we know, there are no previous studies on the discrimination of Tunisian pine seeds through NIR spectroscopy. Accordingly, the aim of this work was to determine the lipid profile and discriminate the seeds of three different pine species (Pinus halepensis, Pinus brutia and Pinus pinaster) using NIR spectroscopy.”

I would suggest to remove this sentence from this paragraph, since referring to materials and methods.

Reply: The authors acknowledge the reviewer comment. The sentence was removed as suggested.

Materials and methods section

Since all all the subsequent analyses (including NIR spectroscopic analysis) were performed on crushed seeds, the authors should add a short introducing paragraph where they explain the preparation steps the seeds were subjected to (e.g., crushing how? were they then subportioned?)

Reply: The authors acknowledge the reviewer comment. Before analysis, seeds were ground in a mill (GM Grindomix 200, Retsh, Haan, Germany). This information was added to the manuscript (Line 103).

In my opinion, the division of the Material and Methods section into 4-level-subsection (2.2.2.1) is too confusing. I would condensate the text and rather stop to a three-level sebsection (e.g., 2.2.2)

Reply: The authors acknowledge the reviewer comment. The text was condensed and changed to a three-level subsection, as suggested.

Please, give information about the selected the fatty acids which were measured.

Reply: The authors acknowledge the reviewer comment. FAMEs were identified by comparison of the retention times with those of the standard mixture Supelco 37 Component FAME Mix (Bellefonte, PA, USA). The individual compounds present in the samples are described in Table 2. This information was added to the manuscript (line 55).

Provide an approximate amount of ground samples which were placed into the borosilicate glass vials and further analysed by NIR.

Reply: The authors acknowledge the reviewer comment. About 1.5 to 2 g of samples were place inside the borosilicate glass vials. This information was added to the manuscript.

Add the integration time used.

Reply: The authors acknowledge the reviewer comment. The instrument used (FTLA 2000, ABB) does uses integration time as a parameter of adjustment as it is based on Fourier transform. This parameter of adjustment is only used in dispersive instruments which is not the case.

When was the background correction performed? the authors should specify if the reference was acquired after a well-defined period of time or after measuring a specific number of samples.

Reply: The authors acknowledge the reviewer comment. The background was performed before the spectral acquisition of pine seeds samples and just once, because the spectral acquisition lasted less around 1h without signal deterioration.

If a total of 20 samples were analyzed, hence 14 spectra (70%) were put into the calibration set and 6 spectra (30%) into the validation set. The authors stated that the data repartition was made by ensuring the same proportion of samples in all groups. My question is: how was this proportion ensured if only 2 samples of Maritime and 2 sample of Brutia were considered? How many samples for each class were hence considered? Please, provide an explanation

Reply: The authors acknowledge the reviewer comment. In that situation, only one sample of each specie was used for calibration and the other for validation. Following your last comment on the results and discussion section about using individual samples replicates the PLS-DA was recalculated using the individual samples replicated. In this case, 4 spectra of maritime pine and 4 spectra of brutia pine were used for calibration and two spectra of each for validation. This information was added to the manuscript and the sentence referring that the average was considered in section 2.3 was removed from the manuscript. The new results were presented, analysed and discussed in the manuscript..

The evaluation of the calibration models based on these parameters was already reported (few lines above). Please, here refer just to RMSEP and R2P

Reply: The authors acknowledge the reviewer comment. The sentence was changed as requested.

Results and discussion section

Apart from literature, the authors should made an attempt to explain the differences in lipid content based on their results and, in particular, based on the characteristics of the samples analyzed, in terms of fraction of variability included into the dataset (related to origin, climatic conditions, maturation stutus, species, etc.)

Reply: The authors acknowledge the reviewer comment. The differences in lipid content are based on the difference of species. This information was added to the manuscript (line 235).

The authors stated that the pine seed were higher in Vitamine E, but it is not clear than what pine seed were higher.

Reply: The authors acknowledge the reviewer comment. The seeds of Aleppo pine presented the higher content of vitamin E. This information was added to the manuscript (line 235).

Please provide a proper bibliographic reference.

Reply: The authors acknowledge the reviewer comment. The reference was added as requested.

Is there any supporting data in literature indicating if the protein or aminoacidic composition of pine seeds vary according to species?

Reply: The authors acknowledge the reviewer comment. Another reference was included showed a difference of content of protein according to the different species studies (line 266).

Please, provide information about the reason why the PC3 was plotted instead of the PC2. Provide also (within the text) the fraction of variability collected by the PC2.

Reply: The authors acknowledge the reviewer comment. PC3 was plotted instead of PC2 as it yielded a better cluster formation according to pine seeds species. This information was added to the manuscript as well the variance captured by PC2 as suggested by the reviewer.

Please, report if this confusion matrix was built usind external validation data or cross-validated (calibration) data.

Reply: The authors acknowledge the reviewer comment. The confusion matrix was obtained using only the validation set. The calibration set was only used to assess the best number of latent variables. This information was added throughout the manuscript to avoid misunderstandings.

Please correct the following sentence: The highest percentage of correct prediction was found when using the combination of spectral region R1 and R3 and with the NIR spectra mean centered.

Reply: The authors acknowledge the reviewer comment. The sentence was corrected as suggested.

In my opinion, the confusion matrices are misleading and difficult to be interpreted if the percentages of correct/wrong classification are reported in relation to the overall number of samples. confusion matrices should be modified and the classification of each sample should be reported in relation to its own class membership (e.g., for the 16 Aleppo seeds, if 10 samples were correctly identified, hence the percentage of correct classification is 62.5% (10/16); if 3 Aleppo samples were wrongly classified as brutia, hence the percentage of wrong classification is 18.8% (3/16).

Reply: The authors acknowledge the reviewer comment. Following the suggestion made by the reviewer regarding the results of the PLS model, the authors decided to use the obtained three NIR spectra for each sample individually instead of considering the average. For the PLS-DA considering the lipid profile chemical values, the same strategy was used (triplicate values for each sample instead of considering the average). Therefore, the PLS-DA models were recalculated and the respective confusion matrices were presented following the suggestions made by the reviewer. Several changes were made in the manuscript to address all this. As can be seen, the obtained results for the PLS-DA models using NIR spectra were similar but the obtained results regarding the reference procedures improved. This can be explained by the increase in the number of samples in the calibration set that allowed the PLS-DA model to better understand the differences between the different pine seeds species. 

This is a too strong statement. The amount of compounds containing CH groups is also strongly influenced by season, geographical area, and climate conditions. The authors have not demonstrated that these sources of variation were taken into account during the sample collection.

Reply: The authors acknowledge the reviewer comment and the sentence was changed as suggested.

Please, clearly report if this confusion matrix was built using external validation data or cross-validated (calibration) data.

Reply: The authors acknowledge the reviewer comment. The confusion matrix was obtained using only the validation set. The calibration set was only used to assess the best number of latent variables. This information was added throughout the manuscript to avoid misunderstandings.

If a very small number of samples can be considered, o some extent, acceptable to develop qualitative discriminant models (as presented above), I think that using chemometric analysis to develop quantitative prediction models calibrated just on 14 samples and validated on 6 samples are not acceptable and no conclusions can be draw about their quality, robustness, stability, reproducibility, and usefulness.

I think that, based on the current research design and sampling plan, these results cannot be scientifically evaluated. I would suggest to remove the paragraph and just focus on discriminant analysis according to pine seed species, using chemical data related to the lipid profile just to support the assertion that the spectral differences related to lipid absorbances were the most influent for the discrimination according to species. One alternative suggestion (even if weak) is to try to recompute the quantitative models by using individual spectral replicates (i.e., by avoiding averaging the spectra) so as to increase the overall number of samples and achieve more robust results.

Reply: The authors acknowledge the reviewer comment and agree with it. Following the reviewer’s suggestion, the authors have recalculated all the PLS models (including the optimization of the best number of LVs) using the three NIR spectra obtained for each sample individually. Overall, the obtained results maintained with no significant changes except the RER parameter that improved in almost all the models due to including samples in the validation set that were located in the extreme of the interval ranges. This is expected taking into consideration the formula of RER parameter. Anyway, these results can be considered more robust than the previous ones because we have a higher number of samples in the calibration and validation set. Several changes were made throughout the manuscript to address this comment.

Conclusions section

Please correct the following sentence: “…and species discrimination in pine seeds.”

Reply: the authors acknowledge the reviewer comment and the sentence was changed as suggested.

Limitations of the applied analytical approaches, with an emphasis of the reproducibility of the methodologies, as well as future perspectives must be discussed in more detail.

Reply: The authors acknowledge the reviewer comment. New sentences were added in the Conclusions section to address the reviewer comment.

Reviewer 2 Report

-    

-        

-          It is difficult to review the manuscript without page and line numbers!

-          In this study, the authors focused on discrimination between spices and quantifying fat content in these pine seeds. However, they took their samples from “different geographic and bio-climatic zones in Tunisia”. As the geographical origin is not studied, so the samples should be taken from the same zone to avoid any interferences of the geographical origin.

-          Another major concern of this study is the limited number of samples studied, especially maritime and brutia pines

-          The resolution of figures 4, 5, 6 and 7 should be improved

Author Response

Reviewer 2 comments

Comments and Suggestions for Authors

It is difficult to review the manuscript without page and line numbers!

Reply: The authors acknowledge the reviewer comment and regret for that situation. The lines and pages were numbered in the revised version.

-In this study, the authors focused on discrimination between spices and quantifying fat content in these pine seeds. However, they took their samples from “different geographic and bio-climatic zones in Tunisia”. As the geographical origin is not studied, so the samples should be taken from the same zone to avoid any interferences of the geographical origin.

Reply: We completely understand the appointment of the reviewer. However, in this case, we collected seeds from different geographical zones because we wanted to characterize and discriminate different species, independently of their origin. So, the aim was to collect the higher number of samples as possible of the three selected species, independently of the geographical region.

-Another major concern of this study is the limited number of samples studied, especially maritime and brutia pines.

Reply: The authors acknowledge the reviewer comment and agree with it. It is true that the number of samples used in this study are low which affects the robustness of the developed work. However, as the tittle refers, this work aimed to demonstrate that is possible to use NIR spectroscopy for the discrimination of different pine seeds species and to quantify the amount of total fats as well as their lipid content as a proof of concept. To attest the robustness of the developed models, more samples would be needed. Several sentences were changed in the results and discussion section to clarify this. Additionally, following the suggestion of other reviewer, the authors decided to recalculate the PLS-DA and PLS models using the three NIR spectra obtained for each sample individually instead of the average. The obtained results were similar than the one showed in the previous version but now more robust due to a high number of samples in the calibration and validation set. Several changes were made throughout the manuscript.

-The resolution of figures 4, 5, 6 and 7 should be improved.

Reply: The authors acknowledge the reviewer comment. The resolution of all the figures was improved to 600 dpi.

Round 2

Reviewer 1 Report

I would like the reviewrs for having addressed all my comments and having provided answers to my question. The quality and readability of manuscript has been improved. 

Author Response

The authors would like to thank the reviewer’s comment. We agree that the quality and readability of the manuscript has been substantially improved due to the reviewer’s suggestions.

Reviewer 2 Report

Abstract:

Line 26-27: The sentence can be replace as follows:

Pine seeds are known for their richness in lipid compounds and other healthy substances. However, the reference procedures that are commonly applied for their analysis are quite laborious, time-consuming, and expensive.

Line 29: please replace “for that purpose,” by “that are”

Line 32: please replace “ascertained” by “investigated”

Line 36: please replace “predictions” by “classification rates”

Line 40: delete the comma “,” after the obtained results

Line 44: Keywords should be different from those in the title. Please check

Introduction:

Line 64-66: it is not clear how the determination of fatty acids by GC will enable the development of other analytical techniques. Please reword or delete

Line 67-72: the authors may improve this paragraph by citing other relevant references dealing with use of spectroscopic techniques for authentication purposes. Examples: https://doi.org/10.3390/foods9081069

https://doi.org/10.1016/j.lwt.2019.01.021

Materials and methods

Line 113: the full name should be given before the abbreviation HPLC when used the first time I the manuscript

Line 115: GC-FID, the same as line 113

Line 193: Aleppo is sometime written with a capital letter and sometimes with a small letter. Please write “Aleppo” throughout the whole manuscript

Results and discussion

Line 275-276: this sentence could be better reworded

The same for the next sentence

Conclusion:

Line 491: replace “besides” by “despite”

Line 492: replace “predictions” by “classification rates”

Line 507: replace “specie” by “species”

Line 508: replace “attest” by “confirm”

Line 514-516: the last sentence can be better reworded. For example: Generally, the primary results obtained using NIR spectroscopy coupled with PLS and PLS-DA suggest that this technique could be applied for the quantification of lipids and the discrimination of pine seeds species.

Author Response

Point-by-point reply_2nd round

Reviewer 2 comments

Comments and Suggestions for Authors

Abstract:

Line 26-27: The sentence can be replaced as follows:

Pine seeds are known for their richness in lipid compounds and other healthy substances. However, the reference procedures that are commonly applied for their analysis are quite laborious, time-consuming, and expensive.

Reply: The authors thank the reviewer comment. The sentence has been changed as suggested.

Line 29: please replace “for that purpose,” by “that are”

Reply: The authors thank the reviewer comment. The sentence has been changed as suggested.

Line 32: please replace “ascertained” by “investigated”

Reply: The authors thank the reviewer comment. The word has been changed as suggested.

Line 36: please replace “predictions” by “classification rates”

Reply: The authors thank the reviewer comment. The sentence has been changed as suggested.

Line 40: delete the comma “,” after the obtained results

Reply: The authors thank the reviewer comment. The comma has been deleted as suggested.

Line 44: Keywords should be different from those in the title. Please check

Reply: The authors acknowledge the reviewer comment. The keywords have been changed as suggested.

Introduction:

Line 64-66: it is not clear how the determination of fatty acids by GC will enable the development of other analytical techniques. Please reword or delete

Reply: The authors acknowledge the reviewer comment. The sentence was changed as suggested.

Line 67-72: the authors may improve this paragraph by citing other relevant references dealing with use of spectroscopic techniques for authentication purposes. Examples: https://doi.org/10.3390/foods9081069

https://doi.org/10.1016/j.lwt.2019.01.021

Reply: The authors acknowledge the reviewer comment. The paragraph has been changed as suggested by the reviewer.

Materials and methods

Line 113: the full name should be given before the abbreviation HPLC when used the first time I the manuscript

Reply: The authors acknowledge the reviewer comment. The full name has been given as suggested.

Line 115: GC-FID, the same as line 113

Reply: The authors acknowledge the reviewer comment. The full name has been given as suggested.

Line 193: Aleppo is sometime written with a capital letter and sometimes with a small letter. Please write “Aleppo” throughout the whole manuscript

Reply: The authors acknowledge the reviewer comment. This was corrected throughout the manuscript and also for Brutia and Maritime.

Results and discussion

Line 275-276:this sentence could be better reworded

The same for the next sentence

Reply: The authors acknowledge the reviewer comment. Both sentences were changed to improve their comprehension.

Conclusion:

Line 491: replace “besides” by “despite”

Reply: The authors acknowledge the reviewer comment. The word was replaced as suggested.

Line 492: replace “predictions” by “classification rates”

Reply: The authors acknowledge the reviewer comment. The words were replaced as suggested.

Line 507: replace “specie” by “species”

Reply: The authors acknowledge the reviewer comment. The word was corrected as suggested.

Line 508: replace “attest” by “confirm”

Reply: The authors acknowledge the reviewer comment. The word was replaced as suggested.

Line 514-516: the last sentence can be better reworded. For example: Generally, the primary results obtained using NIR spectroscopy coupled with PLS and PLS-DA suggest that this technique could be applied for the quantification of lipids and the discrimination of pine seeds species.

Reply: The authors acknowledge the reviewer comment. The sentence has been changed as suggested.

The authors would like to thank all the reviewer’s suggestions that improved the quality of the paper.
